# Adsorption of Blood Components to Extracorporeal Membrane Oxygenation (ECMO) Surfaces in Humans: A Systematic Review

**DOI:** 10.3390/jcm9103272

**Published:** 2020-10-12

**Authors:** Sam Callaghan, Tengyi Cai, Conor McCafferty, Suelyn Van Den Helm, Steve Horton, Graeme MacLaren, Paul Monagle, Vera Ignjatovic

**Affiliations:** 1Department of Haematology Research, Murdoch Children’s Research Institute, MCRI, Parkville 3052, Australia; sam.j.callaghan@gmail.com (S.C.); tengyi.cai@mcri.edu.au (T.C.); conor.mccafferty@mcri.edu.au (C.M.); suelyn.vandenhelm@mcri.edu.au (S.V.D.H.); Steve.Horton@rch.org.au (S.H.); graeme.maclaren@mcri.edu.au (G.M.); Paul.Monagle@rch.org.au (P.M.); 2Department of Paediatrics, The University of Melbourne, Parkville 3052, Australia; 3Department of Cardiac Surgery, The Royal Children’s Hospital, Parkville 3052, Australia; 4Department of Intensive Care, The Royal Children’s Hospital, Parkville 3052, Australia; 5Cardiothoracic Intensive Care Unit, National University Health System, Singapore 119077, Singapore; 6Department of Clinical Haematology, The Royal Children’s Hospital, Parkville 3052, Australia

**Keywords:** ECMO, adsorption, membrane oxygenator, circuit, binding

## Abstract

The accumulation of blood proteins and cells on extracorporeal membrane oxygenation (ECMO) circuits has been proposed as a contributing factor to the coagulopathic state of many patients. This systematic review aims to summarize and discuss the existing knowledge of blood components binding to the ECMO circuits in human patients. A systematic review was conducted using the Medline, PubMed and Embase databases following the Preferred Reporting Items for Systematic Reviews and Meta-Analyses (PRISMA) guidelines. Seven studies were included in this review. Three studies identified a leukocyte adhesion, three studies observed von Willebrand factor accumulation and four studies identified bound platelets on the surface of the circuits. Other identified components included fibrin, albumin, hemoglobin, erythrocytes, progenitor cells, fibronectin and IgG. This systematic review demonstrates the limited state of knowledge when it comes to adsorption to the ECMO circuits in humans. Most of the studies lacked insight or detail into the mechanisms of binding and the interactions between different components bound to the ECMO circuits. Further research is required to comprehensively characterize surface adsorption to ECMO circuits in humans and to define the specific mechanisms of binding, enabling improvements that increase biocompatibility between the blood-circuit interface in this important clinical setting.

## 1. Introduction

Extracorporeal membrane oxygenation (ECMO) is a modified form of cardiopulmonary bypass (CPB), which provides cardiac and/or respiratory support for critically ill patients. ECMO is associated with a high incidence of bleeding and thrombotic events, which may contribute to morbidity and mortality. For example, in one large study [1], 38% of patients on ECMO experienced bleeding and 31% of patients experienced thrombosis, which were associated with decreased survival outcomes of 40% and 33%, respectively.

One of the primary causes hypothesized to initiate a coagulopathic state in ECMO patients is the increased contact of blood components with the artificial ECMO circuit. Blood circulates within an ECMO circuit under different hemodynamic conditions to the patients’ body, altering the shear stress experienced during flow [2] and activating hemostatic components [3], which may adversely affect the patient’s health [1,4,5]. One consequence of ECMO support that has not been investigated in detail is the adsorption of blood components to the surface of the ECMO circuit. Whilst previous studies detailing surface binding in non-ECMO settings [6] and animal models do exist [7,8], limited ECMO circuit binding studies have been conducted in an ex vivo ECMO setting (e.g., circuits obtained from former ECMO patients) in humans. This systematic review aims to summarize the current understanding of cellular and protein adsorption in the setting of ECMO in humans by identifying and examining the studies that have experimentally confirmed adherence of blood components to ECMO circuit surfaces.

## 2. Methods

### 2.1. Study Design

This systematic review was conducted based on the Preferred Reporting Items for Systematic Reviews and Meta-Analyses (PRISMA) guidelines [9].

### 2.2. Search Strategy and Selection Criteria

We searched Medline, Embase and PubMed indexed online databases for studies published from January 1985 to December 2019. The specific search terms used for each database search are detailed in Appendix A. Briefly, studies summarized in this review were identified using the search terms [‘ECMO’ OR ‘membrane oxygenator’] AND [‘adsorption’ OR ‘deposit’] AND [‘blood proteins’ OR ‘blood cells’] as well as using derivatives of these terms. Both veno-venous (VV) and veno-arterial (VA) ECMO were included in the search. Studies were limited to those performed with human samples and those written in the English language. Studies retrieved using these search parameters were screened by two authors (SC and TC) focusing on titles and abstracts for eligibility based on the inclusion and exclusion criteria. Full texts of the chosen articles were assessed by both SC and TC to confirm this.

### 2.3. Inclusion and Exclusion Criteria

Inclusion criteria: (I) ECMO circuit, (II) deposit composition or binding assessed, (III) blood components (cells or proteins) analyzed, (IV) English language and (V) human study.

Exclusion criteria: (I) CPB or other non-ECMO extracorporeal circulation, (II) deposit volume rather than composition assessed, (III) binding of administered drugs analyzed, (IV) study did not investigate deposit composition in the context of ECMO circuit surface and (V) conference abstract.

### 2.4. Data Extraction and Quality Assessment

The included studies were reviewed by VI and PM, with disagreement resolved by discussion. Data extracted included study design, study length, patient age group, ECMO duration and mode, sample types, analysis technique/s and outcomes of any confirmed identification of bound blood components to the ECMO circuits from each study. The risk of bias assessment of each included study was performed in accordance with the Integrated quality Criteria for the Review Of Multiple Study designs (ICROMS) quality assessment tool [10]. All of the studies met the mandatory criteria and minimum score required. The detailed criteria of assessment for each study is presented in Table A1 and the Preferred Reporting Items for Systematic Reviews and Meta-Analyses (PRISMA) 2009 Checklist is presented in Table A2 as part of Appendix B.

## 3. Results

We identified 581 unique studies using our systematic search strategy. Papers were initially screened on their titles and abstracts with the majority of studies (*n* = 565) excluded for not fulfilling the inclusion parameters. Many of the excluded articles were either conducted in the setting of cardiopulmonary bypass circuits (CPB) or in the context of ECMO but focusing on aspects such as the outcomes of anticoagulation strategies or assessing the feasibility of different ECMO circuit designs rather than specifically investigating surface adsorption on ECMO surfaces. Sixteen studies were reviewed for the full text screening with nine studies excluded for either only investigating drug binding to the circuit (*n* = 2), for not analyzing the surface deposition in the context of ECMO circuits (*n* = 2), for being animal studies (*n* = 1) or for being conference abstracts (*n* = 4). Figure 1 shows the PRISMA flow diagram for study inclusion and exclusion of the studies.

### 3.1. Description of Included Studies

A total of seven studies matched the inclusion criteria and are outlined in detail in this systematic review with a summary provided in Table A3 in Appendix B. Six of the seven studies included in this review were ex vivo whilst the seventh study was conducted in an in vitro environment. The seven studies analyzed the adsorption of blood proteins and cells to the blood-circuit interface of the ECMO membrane oxygenators and primarily the polymethylpentane (PMP) oxygenators. Identified plasma proteins included fibrin [11,12], von Willebrand factor (vWF) [13,14,15] and other plasma proteins [13,16] as well as blood cells, which included erythrocytes [11,12], leukocytes [14,15,17], platelets [11,12,15] and progenitor cells [17]. Five of the seven studies utilized immunofluorescent techniques (e.g., flow cytometry, fluorescence microscopy, western blotting) to label specific blood components that adhered to the circuit [11,13,14,15,17]. One study used two-dimensional (2-D) electrophoresis as a method for the identification of specific blood components [16] whilst two studies implemented scanning electron microscopy (SEM) to visualize biomass accumulations. However, in three of the seven studies, the identification of the blood components bound to the ECMO circuit were not the main aim of the studies. Wilm et al. [14] and Steiger et al. [15] observed ECMO oxygenator accumulations as potential predictors for coagulation abnormalities whilst Lehle et al. [11] identified surface deposits as indicators for increased blood flow resistance in ECMO circuits. These articles were therefore limited in terms of the detail that was specified in relation to the ECMO circuit adsorption.

### 3.2. Plasma Protein Binding

#### 3.2.1. Fibrin

Two of the seven studies observed the accumulation of fibrin strands in PMP ECMO oxygenators taken from human patients [11,12]. A third study by Niimi et al. also detected the adhesion of fibrinogen, a precursor for fibrin, to the blood-circuit interface of oxygenators [13].

Lehle et al. and Dornia et al. implemented SEM to directly visualize biomass accumulations in ECMO oxygenators [11,12]. Both studies identified extensive fibrin networks formed on the artificial surfaces with blood cells such as platelets and erythrocytes embedded in the fibrin. Niimi et al. implemented targeted immunofluorescence to identify fibrinogen presence in residue eluted from ECMO oxygenators [13].

#### 3.2.2. Von Willebrand Factor

Three of the seven studies identified the accumulation of vWF in the ECMO oxygenators collected from adult patients following the termination of the ECMO support or replacement of the membrane oxygenator [13,14,15]. Targeted immunofluorescence was performed using monoclonal antibodies to identify vWF deposits on the gas exchange fibers of the membrane oxygenators.

Wilm et al. reported that 11 of 27 PMP oxygenators studied presented with vWF-positive, filament-like structures [14]. Steiger et al. identified accumulation of vWF on all 21 analyzed PMP ECMO oxygenators with high vWF-loading observed in nine circuits [15]. The highest intensity of vWF aggregation was identified at the crossing points of the gas exchange fibers in the oxygenators. The structures of the vWF deposits observed by Steiger et al. were fibrous, cobweb-like, granular or spotty with the type of structures independent of the extent (low or high) of vWF-loading. Niimi et al. also observed vWF adsorption in three different types of oxygenators with the silicone oxygenator adsorbing less vWF than the polypropylene or double polyolefin oxygenators [13].

#### 3.2.3. Other Plasma Proteins

Two of the seven studies analyzed the adsorption of different blood proteins to ECMO oxygenators at the blood-circuit interface.

The study by Owen et al. contained in vitro and ex vivo experiments; however, only the ex vivo component was in the context of ECMO [16]. Circulating plasma protein concentrations in a neonatal patient were analyzed by taking blood samples at different times during the ECMO run (1 h, 17 h, 24 h, 72 h, 96 h and 116 h). Protein residue from the two ECMO oxygenators used by the patient during treatment was eluted and analyzed. Two-dimensional (2-D) electrophoresis was applied to separate the proteins present in the samples for the subsequent identification. There was a consecutive decrease of IgG circulating in the patient’s blood over the course of the ECMO run. Correspondently, the 2-D electrophoresis from the ECMO oxygenator eluate detected a high level of IgG light chains adsorbed to the membrane oxygenator. Albumin and hemoglobin were also identified as dominant blood components present in the eluate removed from the membrane oxygenator, which was estimated due to the inoculation of albumin and the potential lysis of the bound erythrocytes, respectively.

Niimi et al. similarly detected protein adsorption in the ECMO oxygenators [13]. Under in vivo conditions, three different types of ECMO oxygenators (polypropylene, double polyolefin, silicone) were exposed to human whole blood samples. After exposure concluded, the binding materials from the oxygenators were eluted and targeted immunofluorescence was performed on the eluate to characterize the blood proteins that had adhered to the oxygenator surface. Identified blood proteins included vWF, albumin, fibronectin and fibrinogen, which were observed to have the lowest adsorption to the silicone oxygenator compared with the other two oxygenator types.

### 3.3. Blood Cell Binding

#### 3.3.1. Erythrocytes

Lehle et al. and Dornia et al. observed an accumulation of erythrocytes on PMP oxygenator capillaries in an ECMO circuit [11,12]. Lehle et al. performed SEM on 36 dismantled oxygenators whilst Dornia et al. only analyzed one ECMO oxygenator using SEM. Both studies identified networks of fibrin with embedded erythrocytes.

#### 3.3.2. Leukocytes

Three of the seven studies investigated leukocyte adhesion to PMP oxygenators in adults during ECMO support [14,15,17] with one of the studies identifying leukocytoid cells through cells cultivated from PMP oxygenators using flow cytometry [17] and two of the studies visualized cell binding using DAPI (4′,6-diamidino-2-phenylindole) staining [14,15].

Lehle et al. [17] applied CD31 and CD45, two of the leukocytoid cell markers, to identify the presence of leukocytes from cell cultures that had outgrown from the oxygenator eluent. By using the flow cytometry, Lehle et al. detected CD45-positive/CD31-positive leukocytes as the one of the dominant cell types in cell cultures. Wilm et al. [14] also labelled CD45 receptors and marked endothelial vWF-structures to differentiate leukocytes from other nucleated cells (DAPI stained only) by using immunofluorescence microscopy. Wilm et al. characterized the extent of the leukocyte colonizations as (1) uniform, low-density, (2) high-density, particularly around the crossing points of membrane fibers and (3) uniform, high-density pseudomembranous colonies. For regions of low cell density, analysis by fluorescent microscopy identified the majority of adhered nucleated cells in the oxygenators to be CD45-positive leukocytes. These findings were confirmed using flow cytometry. Steiger et al. [15] implemented DAPI staining to identify nucleated cells adjacent to platelet and vWF accumulations with the structures concluded to be platelet-leukocyte aggregates (PLAs). One third of adherent cells on the gas exchange fibers were confirmed to be PLAs with densely populated fibers reporting lower quantities of PLAs. Steiger et al. also fluorescently labelled vWF and platelets within the surface accumulations. High concentrations of the immunofluorescent signals labelling PLA structures were observed to be co-localized in close proximity to the gas exchange fibers, potentially showing evidence for leukocyte adhesion being influenced by/influencing other blood component adsorption to the blood-circuit interface.

#### 3.3.3. Platelets

Four of the seven studies identified the adherence of platelets on the gas exchange surface of the oxygenators used during adult ECMO support [11,12,13,15]. Lehle et al. and Dornia et al. visualized the deposits on the gas exchange membrane using scanning electron microscopy (SEM), identifying platelets embedded in fibrous networks on the PMP oxygenator surface [11,12]. Two of the studies confirmed platelet adherence by using immunofluorescent antibodies to mark platelet receptors, with Steiger et al. labelling receptors for vWF and p-selectin substrates [15] whilst Niimi et al. labelled CD42b and CD61 receptors specifically [13]. Niimi et al. also observed greater platelet accumulation in a double polyolefin oxygenator than in silicone or polypropylene oxygenators. These studies only identified platelets present in accumulations on membrane oxygenator surfaces, not in other ECMO circuit components.

#### 3.3.4. Other Blood Cell Binding

Except for erythrocytes, leukocytes and platelets, other blood cells were also identified accumulated on the surface of PMP ECMO oxygenators by Lehle et al. [17]. The study collected five membrane oxygenators from adult patients at the end of ECMO support or at the time that technical problems arose. Samples taken from adherent cell layers in the oxygenators were subjected to growth factors to promote the proliferation of the collected cells. Immunofluorescence and flow cytometry analysis were used to mark and identify cell receptors such as CD45, CD31, CD90 and CD105 on the cellular growths. Three distinct cell populations cultivated from the membrane oxygenator were identified. Except for the leukocyte population (CD45+/CD31+), which has been mentioned above, an endothelial-like cell population (CD45−/CD31+) and a mesenchymal-like cell population (CD90+/CD105+) were also identified in the cultures. The distribution of these subpopulations depended on the type of membrane oxygenator and cultivation time. The subsequent functional analysis of endothelial-like cells showed the uptake Dil-acetylated low-density lipoprotein and expressed Ulex europaeus I agglutinin, which confirmed the angiogenetic potential of these endothelial progenitor cells.

## 4. Discussion

This systematic review revealed a small pool of studies investigating the characteristics of blood component binding to the ECMO circuit in an ex vivo and in vitro context in humans.

Three studies identified bound components using indirect methods (e.g., 2-D electrophoresis, flow cytometry, fluorescent microscopy), which offered minimal information beyond the presence of particular components [13,16,17]. Although several plasma proteins such as fibrin, fibrinogen, fibronectin, vWF, albumin and hemoglobin were detected [13,16,17], no insight was provided for the order nor mechanism in which the blood components adhered to the artificial surfaces or perhaps influencing other components to bind [3]. The relationship between protein adsorption and subsequent platelet adhesion was discussed by Niimi et al., but these conclusions were inferred in the data according existing knowledge rather than from direct evidence. Extensive studies in non-ECMO specific settings have revealed that fibrinogen is usually the first protein to bind to artificial surfaces [18,19] and vWF was found to bind to fibrinogen layers coated on biomaterials [20] but no current study exists that confirms this in an ECMO context. Establishing temporality of component binding could assist researchers in identifying the proteins that should be targeted when designing new biomaterials or coatings in order to prevent causation from being established for which components are initiating deposit formation.

Though the remaining four studies utilized immunofluorescent microscopy and SEM for direct visualization of the ECMO adsorption [11,12,14,15], as blood cells adhere to the circuit surface by binding to blood proteins already adsorbed to the artificial surfaces [3], observing cellular adsorption still offers little insight into which proteins are initially bound. These studies only superficially observed the binding patterns of thrombotic deposits but lacked detailed investigation into the relationship between protein binding and cell adhesion during thrombotic deposits formation. Only one of the four studies investigated the relationships of these coagulation proteins with platelets during binding in the ECMO circuit [13]. However, these conclusions were made using existing knowledge and were inferred from the data rather than from direct evidence. Studies that have been conducted in a non-ECMO setting showed that the binding of platelets to synthetic biomaterials required pre-adsorbed proteins on the surface such as fibrinogen and vWF [20,21,22]. Therefore, further study is required to confirm a possible causation from protein adsorption influencing platelet adhesion to ECMO circuits.

Four of the seven studies analyzed adhesion during adult ECMO support with only a single neonatal case explored. The findings of these studies are therefore limited to adult ECMO patients and may not translate to pediatric or neonatal patients especially in the context of Developmental Hemostasis. The studies also limit their analysis exclusively to components binding in the membrane oxygenators and do not investigate other circuits such as tubing or pumps, which are known to also accumulate deposits in animal model studies [7,8]. Furthermore, the biomaterials used in ECMO circuits have changed throughout the last thirty years [3,23,24] with the modern studies included in this review exclusively implementing PMP oxygenators compared with the silicone oxygenators preferred in earlier studies. This could implicate that some of the data analyzed in this review is irrelevant to current ECMO oxygenators.

## 5. Conclusions

The binding of blood cells and proteins to ECMO surfaces is likely associated with ECMO complications responsible for high morbidity including thrombosis and bleeding. The current state of knowledge on blood component adsorption to ECMO circuits in humans is limited to a small selection of studies. Several proteins and cells were identified involving ECMO adsorption by either protein identification through sample elution or direct microscopy visualization but none of these extensively characterized the root cause of adherence with the sample restricted to the membrane oxygenator only. Hence, there is a need for future studies. SEM, TEM, confocal microscopy, SWATH-MS and Time-of-Flight Secondary Ion Mass Spectrometry (TOF-SIMS) could be first applied to characterize the different blood components (soluble and cellular elements) involved in the surface adherence during ECMO support and how they interact with the circuit and with each other. In addition, the association between the circuit adsorption and patient demographics, the pathway onto ECMO, the duration of ECMO, clinical events (e.g., bleeding and thrombosis) and therapies such as blood product administration during an ECMO run should be investigated. Once we understand this key knowledge and insight into the mechanism of initiation of the ECMO circuit binding and thrombus formation, we can progress to design circuits that may reduce bleeding and clotting complications during ECMO and improve clinical outcomes on ECMO.

## Figures and Tables

**Figure 1 jcm-09-03272-f001:**
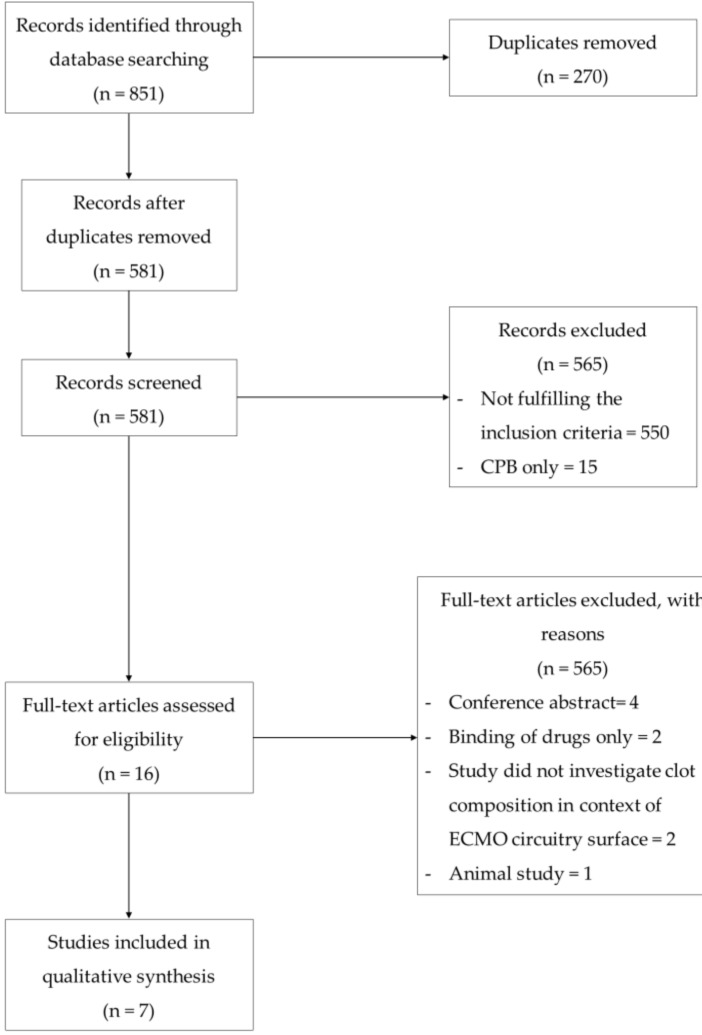
Summary of the study selection process for the systematic review.

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
