# Peer review of "Adsorption of Blood Components to Extracorporeal Membrane Oxygenation (ECMO) Surfaces in Humans: A Systematic Review"

_jcm, 2020, doi:10.3390/jcm9103272_

Round 1

Reviewer 1 Report

I had the opportunity to review the work by Callaghan et al.

They performed a systematic literature review and a qualitative summary of available evidence reporting blood components adsorption into ECMO circuit oxygenator. The topic is of interest, as it investigates the pathophysiological link between blood-circuit surface interaction and coagulative derangements leading to either thrombosis or bleeding in patients with extracorporeal/artificial blood circuits (i.e. “Hemocompatibility”, Circulation 2017 May 23;135(21):2003-2012.).

The systematic search strategy is appropriate and complete. The nature of the studies does not allow for a quantitative synthesis of results, so a narrative-qualitative review is appropriate.

I enclose some comments that may improve the manuscript content:

-Some PRISMA statement elements are lacking: please, add a PRISMA checklist and formal bias assessment (although some items will be missing, and risk of bias will be high due to the nature of included studies).

-Please, explicitly state whether both veno-venous (VV) or veno-arterial (VA) ECMO were included or if search was limited to only one configuration. Please, also add this information in Table 1, relative to each study. This is of interest, as rates of circuit thrombosis (especially pump) seem more common in VV-ECMO (see PLoS One 2020 Jan 27;15(1):e0227793; and Perfusion 2016 Apr;31(3):223-31). Consider to report separate results, if available.

-Please, also add information regarding how many thrombosis/bleeding events were observed in each included study (if available) and if some specific association between clinical outcome and adsorbed blood components was reported. This may add clinical translational value.

-Also, add the reason for circuit removal in each study.

-Results section (lines 108-109): “These articles were therefore limited in terms of the detail that was specified in relation to the ECMO circuit adsorption”. Please be more specific, it is unclear how this lack of data may affect analysis is inclusion criteria are matched.

-Finally, results from the study by Lehle et al [16] are hypothesis-generating. It seems that, with increasing culture time (after 4 weeks) a endothelial (capable of forming capillary-like structures) or mesenchymal phenotype emerged in some specimens: do the Authors believe that this may be a factor associated to endothelization of ECMO components?

Author Response

Reviewer 1

  1. I had the opportunity to review the work by Callaghan et al.

We thank the reviewer for their time and expertise in reviewing our manuscript.

  1. They performed a systematic literature review and a qualitative summary of available evidence reporting blood components adsorption into ECMO circuit oxygenator. The topic is of interest, as it investigates the pathophysiological link between blood-circuit surface interaction and coagulative derangements leading to either thrombosis or bleeding in patients with extracorporeal/artificial blood circuits (i.e. “Hemocompatibility”, Circulation 2017 May 23;135(21):2003-2012.).

We thank the reviewers for their positive feedback.

  1. The systematic search strategy is appropriate and complete. The nature of the studies does not allow for a quantitative synthesis of results, so a narrative-qualitative review is appropriate.

We thank the reviewer for their positive feedback.

  1. I enclose some comments that may improve the manuscript content:

We thank the reviewer for a detailed review of our manuscript.

  1. Some PRISMA statement elements are lacking: please, add a PRISMA checklist and formal bias assessment (although some items will be missing, and risk of bias will be high due to the nature of included studies).

Thank you for your suggestion. The formal bias assessment for the studies included and PRISMA checklist have been added to the manuscript as Table 2 and Table 3 in the Appendix B at the Page of 11 and 13.

  1. Please, explicitly state whether both veno-venous (VV) or veno-arterial (VA) ECMO were included or if search was limited to only one configuration. Please, also add this information in Table 1, relative to each study. This is of interest, as rates of circuit thrombosis (especially pump) seem more common in VV-ECMO (see PLoS One 2020 Jan 27;15(1):e0227793; and Perfusion 2016 Apr;31(3):223-31). Consider to report separate results, if available.

Thank you for your suggestion. We have revised the manuscript to reflect that both VV and VA ECMO were included in the search in “2.2. Search Strategy and Selection Criteria” in method section, and have added a column to Table 1 to specify the ECMO mode relevant for each study.

  1. Please, also add information regarding how many thrombosis/bleeding events were observed in each included study (if available) and if some specific association between clinical outcome and adsorbed blood components was reported. This may add clinical translational value.

Thank you for your suggestion. Unfortunately, the details related to thrombosis/bleeding events for the studies included is not available, as this information is not included in the original studies. Likewise, no specific association between clinical outcome and adsorbed blood components was reported in the studies that were included in our systematic review.

  1. Also, add the reason for circuit removal in each study.

Thank you for your suggestion. We have revised the manuscript and have added a column to the Table 1 to state the reason for circuit removal specific to each study.

  1. Results section (lines 108-109): “These articles were therefore limited in terms of the detail that was specified in relation to the ECMO circuit adsorption”. Please be more specific, it is unclear how this lack of data may affect analysis is inclusion criteria are matched.

We appreciate the reviewer’s opinion. However, we have detailed the limitations of each study in “3.1. Description of included studies” in result section in the manuscript. As stated in the manuscript, on Page 3 and 4, the limitations of these studies are not absence of data, but these studies are not aimed to identify the blood components bound to the ECMO circuit. Despite this, the studies included in our systematic review assessed the deposit composition or binding of the ECMO circuit, with the blood components (cells or proteins) analysed, and were as such relevant and were included the studies into our manuscript.

  1. Finally, results from the study by Lehle et al [16] are hypothesis-generating. It seems that, with increasing culture time (after 4 weeks) a endothelial (capable of forming capillary-like structures) or mesenchymal phenotype emerged in some specimens: do the Authors believe that this may be a factor associated to endothelization of ECMO components?

We believe that accumulation of endothelial-like and mesenchymal-like cells on PMP membranes could be a part of the ECMO binding adsorption. However, the detailed mechanism/s related to the ECMO circuit adsorption remain to be investigated.

We ask you to please review the new version of the manuscript and hope that you accept it for publication in your journal.

Reviewer 2 Report

Authors present systematic review of studies focused on blood products absorption in ECMO membranes. Nowadays, due to lack of evidence it is difficult to fully explain the process of blood products accumulation. Activation of plasma clotting factors, platelet adhesion and aggregation, inflammation and complement are responsible for decreasing membrane oxygenator (MO) performance over time and impending need of MO exchange. I have found no methodological errors. The manuscript is understandable and well written.

From the point of the reader it will be interesting to proposed potential pathways of MO development. Based on authors expertise please add a recommendation regarding future studies to build ideal biocompatible membrane.  What material is most promising? What kind of coating substances should taken account? What kind of experiments should be performed to build material allowing long term use ( in vitro or in vivo)? What parameters would you proposed to observe blood proteins accumulations?  

Author Response

Reviewer 2

  1. Authors present systematic review of studies focused on blood products absorption in ECMO membranes. Nowadays, due to lack of evidence it is difficult to fully explain the process of blood products accumulation. Activation of plasma clotting factors, platelet adhesion and aggregation, inflammation and complement are responsible for decreasing membrane oxygenator (MO) performance over time and impending need of MO exchange. I have found no methodological errors. The manuscript is understandable and well written.

We thank the reviewer for their time and expertise in reviewing our manuscript, as well as their positive feedback. We really appreciate it.

  1. From the point of the reader it will be interesting to proposed potential pathways of MO development. Based on authors expertise please add a recommendation regarding future studies to build ideal biocompatible membrane. What material is most promising? What kind of coating substances should taken account? What kind of experiments should be performed to build material allowing long term use (in vitro or in vivo)? What parameters would you proposed to observe blood proteins accumulations?

We thank the reviewer for their time and expertise in reviewing our manuscript, as well as their positive feedback. We have revised the conclusion section in manuscript at Page 7 to include this recommendation

We ask you to please review the new version of the manuscript and hope that you accept it for publication in your journal